# XMᴇCᴀᴘ: Meme Caption Generation with Sub-Image Adaptability

### Yuyan Chen
chenyuyan21@m.fudan.edu.cn
Shanghai Key Laboratory of Data
Science, School of Computer Science,
Fudan University
Shanghai, China

### Songzhou Yan
szyan21@m.fudan.edu.cn
Shanghai Key Laboratory of Data
Science, School of Computer Science,
Fudan University
Shanghai, China

### Zhihong Zhu
zhihongzhu@stu.pku.edu.cn
Peking University
Beijing, China

### Zhixu Li*
zhixuli@fudan.edu.cn
Shanghai Key Laboratory of Data
Science, School of Computer Science,
Fudan University
Shanghai, China

### Yanghua Xiao*
shawyh@fudan.edu.cn
Shanghai Key Laboratory of Data
Science, School of Computer Science,
Fudan University
Shanghai, China

## ABSTRACT

Humor, deeply rooted in societal meanings and cultural details, poses a unique challenge for machines. While advances have been made in natural language processing, real-world humor often thrives in a multi-modal context, encapsulated distinctively by memes. This paper poses a particular emphasis on the impact of multi-images on meme captioning. After that, we introduce the XMᴇCᴀᴘ framework, a novel approach that adopts supervised fine-tuning and reinforcement learning based on an innovative reward model, which factors in both global and local similarities between visuals and text. Our results, benchmarked against contemporary models, manifest a marked improvement in caption generation for both single-image and multi-image memes, as well as different meme categories. XMᴇCᴀᴘ achieves an average evaluation score of 75.85 for single-image memes and 66.32 for multi-image memes, outperforming the best baseline by 3.71% and 4.82%, respectively. This research not only establishes a new frontier in meme-related studies but also underscores the potential of machines in understanding and generating humor in a multi-modal setting.

## CCS CONCEPTS

• **Computing methodologies → Artificial intelligence**.

## KEYWORDS

Large Multi-modal Models, Meme Caption, Text Generation

## 1 INTRODUCTION

As the rise of social media, memes have become a ubiquitous form of communication and entertainment. A key research task relevant to meme is meme caption generation, which involves creating captions that complements the humor or message of the image to enhance its appeal and shareability [14, 19, 20]. Meme caption generation is valuable for applications and scenarios such as social media content creation, marketing strategies, and digital communication enhancement where engaging visual content is crucial [16, 22, 30, 35, 36, 45, 60].

---

*The corresponding authors.

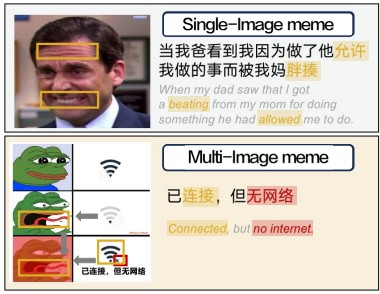

**Figure 1: The distinction between single-image and multi-image memes.**

The existing research efforts on meme caption generation mainly focus on using single-image memes [5, 41, 50, 53, 54]. As the example shown in Fig. 1(a), given the image of a meme featuring a man with a pained expression, the multi-modal model is requested to generate a fitting and humorous caption that aligns with the image's content, such as "When my dad saw that I got a beating from my mom for doing something he had allowed me to do.", indicating the ironic situation where the father permits something that the mother punishes, leading to the man's distressed expression. To achieve this, some work such as Chauhan et al. [8] and Li et al. [32] provide humor datasets from TV and memes respectively, while others such as Ritschel et al. [49] and Ritschel et al. [48] focus on robot humor.

However, all the existing methods can not be adopted when there are multiple sub-images in a meme, each of which probably has different theme and emotion. As the example shown in Fig. 1(b) which is a multi-image meme consisting of three separate sub-images stacked vertically: The first sub-image is a drawing of a character smiling with a good signal Wi-Fi icon. The second sub-image shows the same character looking very upset, with a Wi-Fi icon that has only one bar. In the third sub-image, the character has the same expression as the second one but combining with a red background with an exclamation mark inside the Wi-Fi icon, indicating no internet connection. The caption is "Connected, but no internet.". This multi-image meme tells a story of someone's deteriorating

mood as their internet connectivity worsens, which is relatable to many people's experiences with internet issues. Understanding the relationships between these sub-images and their connection to the overall meme theme poses two challenges. i) *Effective Integration of Composite Information*: In multi-image memes, it's crucial to effectively integrate information from all sub-images to generate a caption that reflects the characteristics of each sub-image while fitting the overall meme context. ii) *Handling the Complexity of Shared Captions*: As the captions for multi-image memes are generally shared across all sub-images, the generated caption needs to maintain consistency between different sub-images while being flexible enough to accommodate the unique content of each sub-image.

Given the absent of benchmark on caption generation for multi-image memes, in this work we construct a new meme datatset (in Chinese) including both single-image and multi-image memes. Specifically, we construct our dataset by sourcing memes from open platforms and meme websites, categorize them by structure (i.e. single-image, multi-image) and emotion (i.e. self-praise, praise of others, self-mockery, and mockery of others), and then balance the collection through downsampling to construct a representative mix of single-image and multi-image memes with varied emotions.

Based on this dataset, we novelly propose XMᴇCᴀᴘ, a new approach which employs separate feature extraction for images and captions, introduces an adaptive transformation to capture the global and token-wise connections between the two, and uses the global and local similarities as supervision signals. We further enhance the LLMs by applying Supervised Fine-Tuning and Reinforcement Learning. Our reward model also incorporates these multi-granularity similarities as a part of the reward signal. Through comprehensive experiments, we demonstrate superior performance in caption generation for both single and multi-image memes.

To summarize, our contributions are in three-fold:

- We recognize the impact of multi-images on meme captioning and offer a novel methodology named XMᴇCᴀᴘ for meme caption generation. XMᴇCᴀᴘ is characterised by supervised fine-tuning and reinforcement learning based on an innovative reward model, which factors in both global and local similarities between visuals and text.
- We conduct extensive experiments to validate the superiority of our approach over current benchmarks in both single and multi-image meme caption generation, alongside promising results in conventional multi-modal humor detection tasks.
- Through visualization analysis, our research further underscores the ability of our proposed XMᴇCᴀᴘ to discern intricate associations between memes and their corresponding captions, setting the stage for advanced meme-related research in the future.

## 2 TASK AND DATASET

The challenge we address is the automatic generation of captions for a diverse array of memes. The problem is two-fold: first, to correctly classify memes into two structural categories (single-image or multi-image) and second, to identify the sentiment of the meme, categorized as self-praise, praise others, self-mockery, or mockery of others. The ultimate goal is to develop an algorithm capable

**Table 1: Number of tokens in meme captions for the entire dataset and various categories, including average, maximum, and minimum counts.**

| Memes | Amount |
|---|---|
| Avg. tokens in captions of memes | 16.6 |
| Max tokens in captions of memes | 25 |
| Min tokens in captions of memes | 5 |
| Avg. tokens in captions of self-praise memes | 14.7 |
| Max tokens in captions of self-praise memes | 21 |
| Min tokens in captions of self-praise in memes | 8 |
| Avg. tokens in captions of praise others in memes | 17.4 |
| Max tokens in captions of praise others in memes | 25 |
| Min tokens in captions of praise others in memes | 10 |
| Avg. tokens in captions of self-mockery in memes | 13.3 |
| Max tokens in captions of self-mockery in memes | 20 |
| Min tokens in captions of self-mockery in memes | 5 |
| Avg. tokens in captions of mock others in memes | 11.5 |
| Max tokens in captions of mock others in memes | 19 |
| Min tokens in captions of mock others in memes | 6 |

of generating a caption that is congruent with both the meme's structure and sentiment, enhancing the humor and communicative intent of the meme.

**Task definition.** The meme caption generation task $G$ aims to produce a caption $\hat{c}_i$ for a given image $m_i$ in a meme. The performance of this task can be measured by a scoring function $S(m_i, c_i, \hat{c}_i)$ which evaluates the generated caption $\hat{c}_i$ against the ground truth $c_i$. The overall goal is to maximize the performance score across all memes in the given dataset as follows:

$$\hat{c}_i = G(m_i), \quad O = \max \sum_{i=1}^{N} S(m_i, c_i, \hat{c}_i). \tag{1}$$

**Dataset construction.** We construct a large-scale meme dataset, comprising 18,110 memes from open-sourced platforms [1] [2] and memes from various Chinese meme images websites. We classify memes based on their structure into single-image memes and multi-image memes. The latter emphasizes connections between individual sub-images alongside the relationship with the caption. We further categorized memes by their sentiment into four types: self-praise, praise others, self-mock, and mock others. To maintain a balanced dataset, we undertook downsampling. The final set contains 12,320 memes with 54% single-image memes and 46% multi-image memes. In terms of sentiment types, we self-praise accounts for 21%, praise others 23%, self-mockery 29%, and mock others 27%. Caption lengths are detailed in Table 1.

## 3 METHODOLOLOGY

In this section, we propose a novel Chinese meme caption generation approach named XMᴇCᴀᴘ, which incorporates visual and textual features through adaptive transformation layer and utilize image-text attention to generate captions. The framework is shown in Fig. 2.

### 3.1 Feature Extraction

Our proposal first separately enhances and extracts features from images and captions. Specifically, we first categorize meme images

---

[1] https://github.com/chineselzf/memeplate

[2] https://github.com/HumorComputing/CCL2021-Humor-Computation

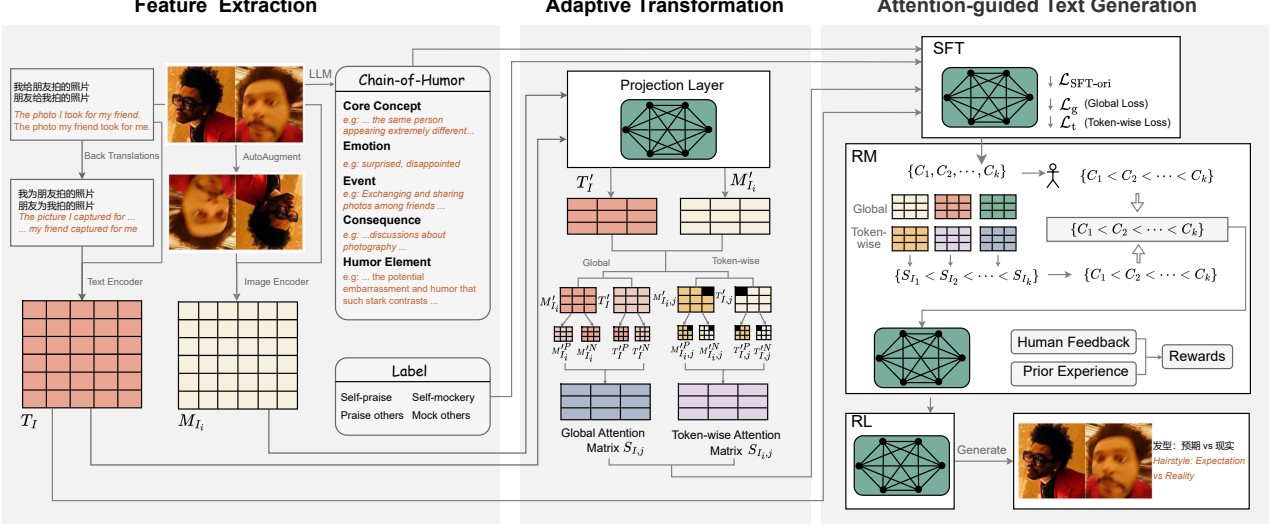

**Figure 2: The overview of the proposed Chinese Meme caption generation framework XMeCap.**

into single-image memes and multi-image memes manually. For multi-image memes, we use OpenCV-Python to precisely identify each sub-image and its boundaries and capture their coordinates by selecting regions of interest (ROIs). This process involves declaring a mouse click function to outline the boundaries of each sub-image. Once these boundaries are established, we apply image enhancement to each sub-image individually. With AutoAugment [23], we transform each original sub-image of an image (denoted as $I$) into enhanced versions. This includes applying techniques such as cropping and rotating, which are customized according to the specific features of each sub-image. For feature extraction, we adopt a powerful large multi-modal model (referred to as LMM, we use LLaVA-1.5-7B here) as a visual encoder to extract deep features from each original and enhanced sub-image. This process allows us to capture unique detailed features in each sub-image, such as the morphology and color gradients of objects. Since the text areas lie outside the boundaries of all sub-images in multi-image memes, we consider the text as a shared caption across all sub-images. The text enhancement process, using back-translation and a Transformer-based LLM (we use Baichuan2-7B here), extracts deep features for these shared texts. These extracted features are then used for each sub-image, considering the shared text as a common caption complementing the entire set of sub-images.

To generate effective meme captions, our proposal generates descriptions for each sub-image based on the LMM. Then, these descriptions are transformed into structured text with a "chain-of-humor" template [15, 21]. For each sub-image, this template includes creating a narrative that encompasses the core concept (such as the main object in the sub-image), emotion (such as surprise), event (such as sharing a photo), consequence (such as discussing the photo), and humor elements (such as using anthropomorphism). Each step of this process, including feature extraction, adaptive transformation, and attention-guided text generation, takes the specific sub-image source (i.e., the index of the sub-image relative

to the meme image) as input, ensuring a tailored approach for both single-image memes and multi-image memes.

## 3.2 Adaptive Transformation

This process aims to effectively merge the image features and caption features of each image area into a unified space [18, 35]. Each feature uses an independent trainable linear layer for projection. This transformation is achieved through trainable weight matrices and corresponding bias terms. Specifically, for each image area (i.e. part of the image and generally not the sub-image) represented as $M_{I_i}$ and its corresponding caption features $T_I$, we apply the following linear transformation:

$$M'_{I_i} = W_{M_{I_i}} \cdot M_{I_i} + b_{M_{I_i}}, \quad T'_I = W_{T_I} \cdot T_I + b_{T_I}, \quad (2)$$

where $W_{M_{I_i}}$ and $W_{T_I}$ are trainable weight matrices, and $b_{M_{I_i}}$ and $b_{T_I}$ represent the respective bias terms.

Next, we use the self-attention mechanism of the LLM to calculate the token-level similarity between each area in the image and each word in the caption, denoted as $S_{I_i}$. Note that the self-attention mechanism embedded in LLMs' architecture sometimes differs from the general self-attention mechanism, such as the ROPE encoding in Baichuan. The entire calculation process is as follows:

$$E_{I_i,j} = (M'_{I_i} W^Q) \cdot (T'_{I_j} W^K), \quad (3)$$

$$S_{I_i,j} = \frac{\exp(E_{I_i,j}/\sqrt{d_k})}{\sum_{u=1}^n \exp(E_{I_i,u}/\sqrt{d_k})}, \quad (4)$$

where $W^Q$ and $W^K$ are the weight matrices of queries and keys in the self-attention mechanism, $d_k$ is the dimension of the key vectors, which adjusts the scale of the dot product, $E_{I_i,j}$ is the unnormalized attention weight, and $S_{I_i,j}$ is the normalized attention weight representing the similarity between the $i$-th image area and the $j$-th caption word. To compute the global attention, denoted as $S_{I,j}$, we average the token-level attention across the image and

caption to obtain a global view. The computation is as follows:

$$S_{I,j} = \frac{1}{N} \sum_{i=1}^{N} S_{I_i,j}. \tag{5}$$

Finally, we use the global attention and token-level attention to construct a loss function aimed at maximizing the similarity between images and captions while minimizing the similarity of unrelated combinations. This is achieved through contrastive loss, computed as follows:

$$L_I = -\log \frac{\exp(S_{I_i,j}/\tau)}{\sum_{k=1}^{N'} \exp(S_{I_i,k}/\tau)}, \tag{6}$$

where $S_{I_i,j}$ is the similarity score between the $i$-th image area and the $j$-th caption word, $S_{I_i,j}$ is the similarity score between the $i$-th image area and all words in a caption (including matching and non-matching, denoted as $k$), $N'$ is the number of caption words, and $\tau$ is the temperature parameter, which adjusts the sensitivity of the loss function.

## 3.3 Attention-guided Text Generation

Given the prior knowledge provided by the aforementioned global and token-level similarities representing the relevance between images and captions, our proposal adopt attention-guided text generation method including Supervised Fine-Tuning (SFT) and Reinforcement Learning (RL) to improve the performance of generating meme captions [11, 13, 43, 44]. First is the SFT step, where the LLM aims to generate meme captions that are close to ground-truth captions. A key aspect of this process is to ensure that the LLM's predicted similarity aligns well with the similarity derived from prior knowledge (global and token-level). To this end, we introduce an additional loss component that utilizes the Kullback-Leibler (KL) divergence to measure the difference between these similarities, as shown below:

$$p_{S_I} = \frac{\exp(S_I)}{\exp(S_I) + \exp(S^{\text{SFT}})}, \quad q_{S_I} = \frac{\exp(S^{\text{SFT}})}{\exp(S_I) + \exp(S^{\text{SFT}})}, \tag{7}$$

$$p_{S_{I_i,j}} = \frac{\exp(S_{I_i,j})}{\exp(S_{I_i,j}) + \exp(S_{I_i,j}^{\text{SFT}})}, \quad q_{S_{I_i,j}} = \frac{\exp(S_{I_i,j}^{\text{SFT}})}{\exp(S_{I_i,j}) + \exp(S_{I_i,j}^{\text{SFT}})}, \tag{8}$$

$$\mathcal{L}_g = \lambda_g \text{KL}(p_{S_I} || q_{S_I}), \quad \mathcal{L}_t = \lambda_t \sum_{i,j} \text{KL}(p_{S_{I_i,j}} || q_{S_{I_i,j}}), \tag{9}$$

where $S^{\text{SFT}}$ denotes the similarity score predicted by the LLM for the image and its caption. Specifically, it represents the cosine similarity between the captions generated through Supervised Fine-Tuning (SFT) and the ground-truth captions. $S_I$ refers to the similarity score between the image and its caption, calculated based on prior knowledge, including global and token-level similarities. $S_{I_i,j}^{\text{SFT}}$ indicates the similarity score between the $i$-th image area and the $j$-th word in the caption as predicted by the LLM. $S_{I_i,j}$ is the similarity score for captions obtained through SFT, which is determined by the cosine similarity between captions generated through SFT and the ground-truth captions. This score represents the similarity between the $i$-th image area and the $j$-th caption word, based on prior knowledge such as global and token-level similarities. Here, the total loss for SFT, including the original SFT loss and the introduced loss based on prior knowledge similarities at the global and token levels, becomes:

$$\mathcal{L}_{\text{SFT}} = \lambda_{\text{SFT-ori}} \mathcal{L}_{\text{SFT-ori}} + \lambda_g \mathcal{L}_g + \lambda_t \mathcal{L}_t, \tag{10}$$

where $\lambda_{\text{SFT-ori}}$, $\lambda_g$ and $\lambda_t$ is a trainable weight.

The next step involves building a reward model to align captions with human preferences. The reward model links the generated caption $y$ with the ground truth one $t$ to calculate a reward $r_i = R_i(y, t)$. In detail, we initially manually evaluate 1% of the captions generated during the SFT process with the criteria shown in Table 2. Here, the annotators are three volunteers. They kindly provide help without any compensation. They rank according to the scoring criteria mentioned in the response to W21. These individuals have not seen the ground truth in advance, and the order is only accepted if the agreement exceeds 0.7. This process results in a ranked sequence, represented as $\{c_1, c_2, \ldots, c_{k-1}, c_k\}$. Based on the calculated rewards, we construct ordered sequences for captions on each image, such as $\{c_1 < c_2 < \ldots < c_{k-1} < c_k\}$, where a larger reward indicates a better match between $y$ and $t$. During caption generation, as tokens emerge, we compute their attention towards specific image areas. If a token's attention closely matches prior attention distribution, it receives a reward; if not, it's given a penalty. This process yields a new ranking. By balancing human evaluations with attention-based rankings, we determine the final sequence for captions generated in the SFT process. To train the reward model, we also employ "Baichuan2-7B" as the backbone, modifying its softmax layer to a linear one. This reward model takes a caption and outputs a score which represents the caption's quality. We collate captions from the final ranking sequence and apply the Pairwise Ranking Loss, depicted as follows:

$$L_r = -\frac{1}{\binom{k}{2}} E_{(x,y_w,y_l) \sim D} \left[ log(\sigma(r_\theta(x, y_w) - r_\theta(x, y_l))) \right], \tag{11}$$

where $x$ symbolizes the original caption. $y_w$ and $y_l$ are indicative of captions with higher and lower scores in the given ranking pair, respectively. $r_\theta$ is the scalar output from the reward model, with $D$ being the set of ranking pairs, and $k$ representing the count of captions produced during the SFT process. This reward model's efficacy lies in its capacity to attribute higher scores (rewards) to superior captions and lower scores (penalties) to inferior captions, adeptly mirroring human preferences and LLM's preference.

After that, we feed captions $c$ generated by the SFT model into the RL model $\pi_\phi^{RL}$ to obtain a more human-preferred captions $y$. We first input $(x, y)$ into the reward model $r_\theta$ and calculate a score (i.e., reward), which represents the real-time feedback from the reward model. Next, we aims to maintain similarity between the RL model and the SFT model with Kullback-Leibler (KL) divergence. Finally, we combine the two loss functions as follows:

$$L_\phi^{r_\theta} = E(x, y) \sim D_{\pi_\phi^{RL}} [r_\theta(x, y)], \tag{12}$$

$$L_\phi^{\text{SFT}} = -log(\pi_\phi^{RL}(y|x)/\pi^{SFT}(y|x)), \tag{13}$$

$$L_{\text{RL}} = w_1^{RL} \cdot L_\phi^{r_\theta} + w_2^{RL} \cdot L_\phi^{\text{SFT}}, \tag{14}$$

where $\pi_\phi^{RL}(y|x)$ and $\pi^{SFT}(y|x)$ represent captions generated by RL model and the SFT model, respectively, $w_1^{RL}$ and $w_2^{RL}$ are trainable weights.

## 4 EXPERIMENTS

In this section, we evaluate the performance of our proposed XMe-Cap framework on memes caption generation dataset, including single-image memes and multi-image memes.

**Table 2: The criteria of human evaluation for captions generated after SFT.**

| Metric | Criteria |
|---|---|
| Informativeness | Score 1 - Not Informative: The meme caption is completely uninformative, lacking relevant content that provides context or clarity to the image. |
| | Score 2 - Slightly Informative: The meme caption is slightly informative, offering minimal context or clarification that marginally enhances the image. |
| | Score 3 - Moderately Informative: The meme caption is moderately informative, providing a fair amount of context or clarification that adds some meaning to the image. |
| | Score 4 - Very Informative: The meme caption is very informative, delivering substantial context or clarification that greatly enhances the understanding of the image. |
| | Score 5 - Exceptionally Informative: The meme caption is exceptionally informative, presenting comprehensive context or clarification that significantly enriches the image and user experience. |
| Relevance | Score 1 - Not Relevant: The meme caption is completely unrelated to the image, failing to add any meaningful context or humor related to the image. |
| | Score 2 - Slightly Relevant: The meme caption is slightly relevant, providing little context or humor that connects with the image. |
| | Score 3 - Moderately Relevant: The meme caption is moderately relevant, offering a moderate connection to the image with some contextual humor or meaning. |
| | Score 4 - Very Relevant: The meme caption is very relevant, adding significant context or humor that strongly connects with the image. |
| | Score 5 - Exceptionally Relevant: The meme caption is exceptionally relevant, perfectly complementing the image with highly pertinent context or humor, enhancing the overall impact. |
| Creativity | Score 1 - Not Creative: The caption is completely uncreative, lacking originality and wit. |
| | Score 2 - Slightly Creative: The caption is slightly creative, offering basic and predictable humor. |
| | Score 3 - Moderately Creative: The caption is moderately creative, presenting a conventional yet slightly innovative twist. |
| | Score 4 - Very Creative: The caption is very creative, featuring a unique and clever idea or joke. |
| | Score 5 - Exceptionally Creative: The caption is exceptionally creative, displaying high originality and a sharp, memorable wit. |
| Humorous | Score 1 - Not Humorous: The caption is completely not humorous, failing to evoke any laughter or amusement. |
| | Score 2 - Slightly Humorous: The caption is slightly humorous, eliciting a mild smile or light chuckle at best. |
| | Score 3 - Moderately Humorous: The caption is moderately humorous, generating a genuine laugh or amusement with its content. |
| | Score 4 - Very Humorous: The caption is very humorous, provoking strong laughter and enjoyment with its clever humor. |
| | Score 5 - Exceptionally Humorous: The caption is exceptionally humorous, delivering a memorable and hilarious experience that leaves a lasting impression of amusement. |

## 4.1 Experimental Setups

We conduct our experiments on four Nvidia A100 GPUs, each with 80GB of memory, using PyTorch in Python. For enhanced training efficiency, we utilize DeepSpeed. We set the maximum sequence length for both input and output sequences to 1024 tokens. The training process is set to 20 epochs. $\lambda_{\text{SFT-ori}}$, $\lambda_g$, and $\lambda_t$, which control the weight of original SFT process, the weight of global similarity loss ($\mathcal{L}_g$), and the weight of token-wise similarity loss ($\mathcal{L}_t$), in the total loss, are initially set to 0.4, 0.2, and 0.4, respectively. $w_1^{RL}$ and $w_2^{RL}$, which adjust the weight of the reward model loss, as well as the weight of the loss for maintaining similarity between the RL model and the SFT model, are initially set to 0.4 and 0.6, respectively.

Specifically, for $\lambda_{\text{SFT-ori}}$, $\lambda_g$, and $\lambda_t$, we believe SFT is more important because fine-tuning has been proven to be more effective in improving the performance of downstream tasks and is supervised. Since we are capturing the relationship between each sub-image and the caption, we consider the token level to be more important. Therefore, the order of the values is $\lambda_{\text{SFT-ori}} \geq \lambda_t \geq \lambda_g$. We try different combinations using prior knowledge, with $\lambda_{\text{SFT-ori}}$, $\lambda_t$, and $\lambda_g$ being 0.4, 0.4, and 0.2; 0.5, 0.3, and 0.2; and 0.6, 0.2, and 0.2, respectively. For $w_1^{RL}$ and $w_2^{RL}$, we use prior knowledge to try different combinations, including 0.5 and 0.5, 0.4 and 0.6, and 0.6 and 0.4. Each experiment is trained with these sets of parameters, and the best-performing set was chosen. Each baseline is optimized through SFT and RL with the training set. The loss function of SFT for these baselines is only $\mathcal{L}_{\text{SFT-ori}}$ while that for XMeCap is $\lambda_{\text{SFT-ori}}$, $\lambda_g$ and $\lambda_t$.

Due to many LMMs being pre-trained on English corpora, to minimize the impact of language on model performance while preserving the characteristics of Chinese memes, we utilize Tencent Cloud's API [3] to translate Chinese captions for training and evaluation. Ultimately, the LMMs output results in English, which we then translate back into Chinese for presentation.

## 4.2 Datasets, Baselines and Metrics

The selected datasets contain UR-FUNNY [27] which is designed for humor understanding, MUStARD [6] which focuses on multimodal sarcasm detection, MHD [47] which predicts laughter in sitcoms

[3]https://cloud.tencent.com/

**Table 3: The performance of XMeCap in comparison to other baselines in Meme caption generation on the single-image memes.**

| (Single) | Info | Rele | Crea | Humo | HAverage | BLEU | ROUGE | CIDEr | METEOR | MAverage | Average |
|---|---|---|---|---|---|---|---|---|---|---|---|
| | | | Human | | | | | Automatic | | | |
| *PLM* | | | | | | | | | | | |
| S2S [54] | 38.20 | 21.32 | 22.56 | 17.34 | 24.86 | 18.78 | 36.33 | 55.74 | 17.81 | 32.17 | 28.51 |
| Dank Learning [2] | 39.94 | 23.46 | 27.72 | 24.76 | 28.97 | 23.76 | 43.18 | 62.57 | 23.58 | 38.27 | 33.62 |
| Transformer [50] | 46.67 | 28.45 | 33.06 | 26.55 | 33.68 | 30.11 | 50.00 | 63.09 | 32.86 | 44.02 | 38.85 |
| MEMEIFY [53] | 49.25 | 32.13 | 40.18 | 29.37 | 37.73 | 32.54 | 53.35 | 69.21 | 37.11 | 48.05 | 42.89 |
| *LMM* | | | | | | | | | | | |
| BLIP-2-7B [29] | 60.25 | 43.67 | 49.22 | 39.26 | 48.10 | 48.22 | 74.02 | 85.28 | 50.17 | 64.42 | 56.26 |
| MiniGPT-4-7B [59] | 61.08 | 46.31 | 51.08 | 40.22 | 49.67 | 50.02 | 75.31 | 87.44 | 52.18 | 66.24 | 57.96 |
| InstructBLIP-7B [24] | 62.55 | 48.46 | 55.33 | 42.65 | 52.25 | 52.36 | 79.77 | 88.13 | 53.49 | 68.44 | 60.34 |
| LLaVA-7B [38] | 61.37 | 47.44 | 55.32 | 43.07 | 51.80 | 54.26 | 78.12 | 88.33 | 54.92 | 68.91 | 60.35 |
| Unified-IOXL-2B [42] | 72.75 | 60.32 | 57.22 | 48.08 | 59.59 | 56.72 | 85.34 | 90.56 | 57.24 | 72.47 | 66.03 |
| Shikra-7B [9] | 76.36 | 66.58 | 58.13 | 52.57 | 63.41 | 57.05 | 88.76 | 91.87 | 58.38 | 74.02 | 68.71 |
| Qwen-VL-Chat-7B [3] | 79.62 | 68.36 | 58.14 | 53.88 | 65.00 | 57.19 | 88.90 | 91.98 | 59.17 | 74.31 | 69.66 |
| LLaVA-1.5-7B [37] | 80.10 | 69.34 | 58.23 | 54.01 | 65.42 | 57.21 | 89.33 | 92.14 | 59.25 | 74.48 | 69.95 |
| GPT4v [4] | 80.28 | 70.42 | 59.16 | 55.83 | 66.42 | 57.33 | 91.94 | 93.33 | 60.08 | 75.67 | 71.05 |
| **XMeCap** | **83.58** | **76.77** | **63.82** | **61.17** | **71.34** | **62.98** | **94.87** | **97.26** | **66.31** | **80.36** | **75.85** |
| ↑(%) | 1.38 | 4.32 | 5.78 | 5.14 | 3.93 | 4.95 | 2.72 | 2.38 | 4.99 | 3.51 | 3.71 |

based on multimodal data. All results are reported on the 20% subset split from the original dataset. The remaining 80% subset is regarded as the training set. Specifically, we separately compare the results on single-image memes and multi-image memes. The prompt template for generating captions with baselines is "What is a humorous short sentence that complements the image as a meme?".

We categorize our evaluation metrics into two groups: meme-specific metrics and general metrics. Meme-specific metrics contain informativeness, relevance, creativity, and humorous, measuring the human-like quality of the generated captions. General metrics contain BLEU [46], ROUGE [34], CIDEr [55], and METEOR [4], measuring the relevance and diversity of the generated captions. The rating scale of meme-specific metrics are all from 1 to 5, where 1 means the worst and 5 means the best. The final scores will be scaled to 1-100. We enroll three volunteers, and each of them is required to give scores for the randomly selected 1000 memes with generated captions. We also calculate Inter-rater agreement of Krippendorff's Alpha (IRA) to ensure the confidence of human ratings. For the controversial ratings which have low agreements (<0.7), we discard this caption.

## 4.3 Main Results

Based on the presented experimental results in Table 3 and Table 4, our method XMeCap exhibits impressive performance in

**Table 4: The performance of XMeCap in comparison to other baselines in Meme caption generation on the multi-image memes.**

| (Multi) | Human | | | | | Automatic | | | | | Average |
|---|---|---|---|---|---|---|---|---|---|---|---|
| | Info | Rele | Crea | Humo | HAverage | BLEU | ROUGE | CIDEr | METEOR | MAverage | |
| PLM | | | | | | | | | | | |
| S2S [54] | 35.21 | 18.78 | 14.36 | 10.15 | 19.63 | 11.69 | 28.31 | 46.23 | 10.34 | 24.14 | 21.88 |
| Dank Learning [2] | 40.54 | 23.26 | 22.55 | 19.32 | 26.42 | 18.91 | 34.25 | 55.84 | 13.69 | 30.67 | 28.55 |
| Transformer [50] | 48.57 | 31.66 | 24.79 | 21.08 | 31.53 | 22.46 | 39.03 | 58.56 | 19.31 | 34.84 | 33.18 |
| MEMEIFY [53] | 51.28 | 39.84 | 33.80 | 24.09 | 37.25 | 26.58 | 44.24 | 66.15 | 21.05 | 39.51 | 38.38 |
| LMM | | | | | | | | | | | |
| BLIP-2-7B [29] | 65.13 | 51.43 | 44.03 | 34.23 | 48.71 | 42.79 | 64.58 | 84.28 | 38.11 | 57.44 | 53.07 |
| MiniGPT-4-7B [59] | 65.12 | 51.44 | 45.35 | 35.25 | 49.29 | 43.66 | 66.32 | 85.10 | 39.55 | 58.66 | 53.97 |
| InstructBLIP-7B [24] | 67.98 | 55.66 | 49.04 | 39.43 | 53.03 | 46.77 | 68.03 | 85.74 | 42.13 | 60.67 | 56.85 |
| LLaVA-7B [38] | 68.22 | 55.11 | 48.37 | 40.46 | 53.04 | 47.02 | 69.44 | 86.47 | 41.97 | 61.23 | 57.13 |
| Unified-IOXL-2B [42] | 69.35 | 57.68 | 49.55 | 43.65 | 55.06 | 48.34 | 70.56 | 87.21 | 44.57 | 62.67 | 58.86 |
| Shikra-7B [9] | 69.99 | 59.32 | 49.91 | 45.66 | 56.22 | 48.55 | 71.88 | 87.26 | 46.24 | 63.48 | 59.85 |
| Qwen-VL-Chat-7B [3] | 70.76 | 60.47 | 49.22 | 46.14 | 56.65 | 49.10 | 72.90 | 87.16 | 46.32 | 63.87 | 60.26 |
| LLaVA-1.5-7B [37] | 71.00 | 60.03 | 49.76 | 46.99 | 56.95 | 49.13 | 73.11 | 87.32 | 46.87 | 64.11 | 60.53 |
| GPT4v [5] | 71.08 | 60.55 | 50.13 | 47.14 | 57.23 | 50.26 | 73.36 | 88.30 | 47.92 | 64.96 | 61.09 |
| **XMeCap** | **76.42** | **65.77** | **55.92** | **52.49** | **62.65** | **56.62** | **78.11** | **93.24** | **52.02** | **70.00** | **66.32** |
| ↑(%) | 3.59 | 5.91 | 4.94 | 8.12 | 5.42 | 6.77 | 3.86 | 3.59 | 3.58 | 4.29 | 4.82 |

**Table 5: The contributions of each component of XMeCap in single-image memes.**

| Variants | Info | Rele | Crea | Humo | BLEU | ROUGE | CIDEr | METEOR |
|---|---|---|---|---|---|---|---|---|
| w/o IA | 81.34 | 74.23 | 60.25 | 58.87 | 58.82 | 92.01 | 94.88 | 63.86 |
| w/o TA | 82.56 | 75.82 | 62.31 | 60.02 | 60.98 | 93.11 | 95.89 | 64.31 |
| w/o COH | 74.24 | 68.12 | 53.36 | 53.76 | 52.16 | 86.58 | 89.23 | 56.33 |
| w/o A | 69.45 | 63.57 | 48.55 | 47.36 | 47.13 | 80.22 | 83.64 | 51.02 |
| w/o CL | 79.92 | 72.28 | 58.13 | 57.02 | 56.67 | 90.26 | 93.10 | 61.45 |
| w/o RL | 76.52 | 70.02 | 55.34 | 55.87 | 54.12 | 88.02 | 91.20 | 58.70 |
| XMeCap | 83.58 | 76.77 | 63.82 | 61.17 | 62.98 | 94.87 | 97.26 | 66.31 |

**Table 6: The contributions of each component of our proposed XMeCap in multi-image memes.**

| Variants | Info | Rele | Crea | Humo | BLEU | ROUGE | CIDEr | METEOR |
|---|---|---|---|---|---|---|---|---|
| w/o IA | 72.11 | 59.35 | 53.96 | 47.84 | 52.62 | 74.70 | 91.36 | 50.44 |
| w/o TA | 74.10 | 63.66 | 54.50 | 49.21 | 54.22 | 75.93 | 92.37 | 51.68 |
| w/o COH | 61.77 | 47.43 | 45.62 | 32.14 | 38.44 | 61.89 | 82.39 | 41.30 |
| w/o A | 52.30 | 40.66 | 39.36 | 23.84 | 27.55 | 55.98 | 78.13 | 34.82 |
| w/o CL | 68.36 | 56.37 | 53.77 | 44.76 | 49.29 | 72.26 | 90.35 | 50.16 |
| w/o RL | 65.22 | 53.58 | 51.28 | 38.82 | 43.27 | 66.39 | 87.51 | 47.10 |
| XMeCap | 76.42 | 65.77 | 55.92 | 52.49 | 56.62 | 78.11 | 93.24 | 52.02 |

**Table 7: The average performance of XMeCap on different types of single-image memes and multi-image memes, respectively. We make down-sampling to ensure the equal amount of each category for fair comparison.**

| | Single | | Average | Multi | | Average |
|---|---|---|---|---|---|---|
| | Human | Automatic | | Human | Automatic | |
| Self-praise | 70.33 | 79.65 | 74.99 | 60.44 | 70.05 | 65.25 |
| Praise others | 71.26 | 78.03 | 74.65 | 60.82 | 68.52 | 64.67 |
| Self-mockery | 71.19 | 80.92 | 76.06 | 63.46 | 71.19 | 67.33 |
| Mock others | 72.03 | 82.25 | 77.14 | 65.32 | 72.14 | 68.73 |

**Table 8: The performance of XMeCap in comparison to other baselines in modal-humor detection.**

| | UR-FUNNY | MUStARD | MHD |
|---|---|---|---|
| | Accuracy | Accuracy | Accuracy/F1 score/ROC |
| VTFM | - | - | 68.48/79.12/0.60 |
| MSAM | - | - | 72.37/81.32/0.68 |
| HKT | 77.36 | 79.41 | - |
| **XMeCap** | **91.36** | **94.88** | **90.12/95.01/0.81** |

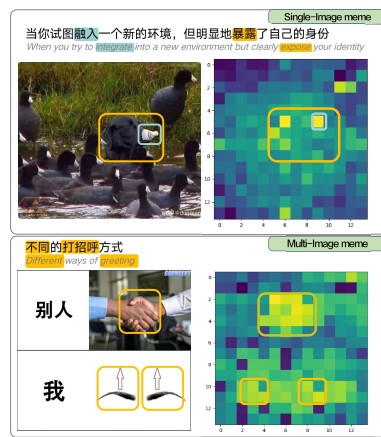

**Figure 3: Illustrative interpretation of meme-caption associations in single-image and multi-image memes.**

both single-image and multi-image meme caption generation. Compared to existing baselines, XMeCap consistently achieves higher scores across various metrics, demonstrating its efficacy. Notably, its performance is closely aligned with that of GPT4, suggesting that our method is competitive and can rival state-of-the-art models in this domain. Moreover, we test 500 samples with GPT-4o, using the same dimensions as human evaluation, and each sample is tested three times. We only use a score if at least two out of three results are the same. We find that the correlation between GPT-4o scores and human scores is very high, reaching 0.935, which indicates that human evaluation can partly be replaced with GPT-4o scoring, thereby reducing manual efforts. From the results presented in Table 7, it's evident that our method XMeCap exhibits varied performance across different meme types, both in single-image and multi-image formats. Interestingly, XMeCap performs notably better in memes with negative connotations, such as "Self-mock" and "Mock others", compared to those with positive themes like "Self-praise" and "Praise others". This suggests that XMeCap has seemingly captured the contrastive nature inherent in memes, finding a more conducive space for caption generation in images with a negative nuance. This insight underscores XMeCap's sub-image adaptability of meme sentiment. The results from Table 8 highlight the performance of our method, XMeCap, after being fine-tuned for modal-humor detection tasks. It is evident that XMeCap achieves commendable results across multiple modal-humor detection benchmarks, exhibiting performance closely rivaling that of GPT4. This underscores the generalization capabilities of our approach.

## 4.4 Ablation Study

Take multi-image meme caption generation as an example, we evaluate the impact of each component in our proposed XMeCap for single-image and multi-image memes as shown in Table 5 and Table 6, respectively. Specifically, w/o IA, Without Image Augmentation, refers to not applying image enhancement techniques to the original image during the feature extraction process. This means

**Table 9: Some high-quality captions of single-image memes generated by the proposed XMᴇCᴀᴘ in comparison to other baselines of meme caption generation.**

| Meme | | | | |
|---|---|---|---|---|
| Style | Self-praise | Praise others | Self-mockery | Mock others |
| Ground truth | When friends from different circles gather together, it's you. | The good friend who always accompanies you when you're feeling down. | The remote control is watching me as I search for it like an idiot. | When the song you chose is up next, but the current person with the microphone hasn't planned on letting go. |
| S2S | Even the sponge is applauding for me. | The fluffy hand of a cat. | The long cat. | I can't believe it. |
| Dank Learning | The underwater party. | The cat's masseuse. | I'm like a cat. | Its gaze is sharper. |
| Transformer | The world's happiness is all because of me. | The benefactor of the pet world. | My laziness is like a cat's. | It's as if I've seen the look of your secret. |
| MEMEIFY | Every party, I'm the brightest star. | Who says humans and animals are different? Just watch their fingertips communicate. | My dream working state. | Surely saw through you again. |
| BLIP-2-7B | SpongeBob SquarePants. | The orange cat puts its paws on a person's hand. | Black and white cat lying on the couch. | Brown and white dog stand in the middle of the stones. |
| MiniGPT-4-7B | It's a funny image. | Looks like someone is playing with the kitten's paws! | I'm just a cat who wants to rest on the couch. | The little one is looking for a new home and is looking forward to a permanent home. |
| InstructBLIP-7B | SpongeBob SquarePants. | I'm not a cat person, I'm a paw-son person | Meow. | The dog is in the middle of the stones, difficult to tell. |
| LLaVA-7B | Funny starfish family photos where they look like a giant starfish. | Cats are awesome and can pop their nails! | Cat, cat, have you drifted on a blue boat? | The dog looks like he's laughing at you, you know? |
| Unified-IOXL-2B | SpongeBob SquarePants is funny. | Cat. | This cat is so long. | The sharp-eyed dog is looking at you. |
| Shikra-7B | SpongeBob SquarePants is an alien. | Those cats who sit on the edge of the window and watch you together. | A cat that looks stretched. | You got it. |
| Qwen-VL-Chat-7B | SpongeBob SquarePants family photo! | Let's shake the paws, kitten | Lying corpse cat | When you want to take a landscape photo, but your dog has a different idea. |
| LLaVA1.5-7B | Spongebob's family is getting bigger and bigger! | The cat's nails are soft, as if they were gently stroked with their fingers. | Cats are lazy, they will bend their bodies on the sofa and look like a long leg. | Don't sleep, I'm sleeping! |
| GPT4v | Looks like someone's ready for a jellyfishing adventure! | When you try to do chores, but your cat has other plans. | Long day, even longer cat. | This disguise is really genius, even I can't see myself! |
| **XMᴇCᴀᴘ** | **Me, the king of the party** | **The moment he touched me, I felt happiness.** | **Where there's a sofa, there I am.** | **Did you arrive late again?** |

**Table 10: Some high-quality captions of multi-image memes generated by the proposed XMᴇCᴀᴘ in comparison to other baselines of meme caption generation.**

| Meme | | | | |
|---|---|---|---|---|
| Style | Self-praise | Praise others | Self-mockery | Mock others |
| Ground truth | When life tries to stop you from moving forward, but you still keep pushing on. | Once the best of friends, always the best of friends. | Every now and then, it's me. | Smokers in their 20s. Smokers in their 30s. |
| S2S | The wind comes, and I still stand. | That dog looks so good in the water. | I'm here, I forgot why. | Strong cat when young. |
| Dank Learning | My hair in the wind. | Look at that dog, is it like a fish? | My mind is blank. | Strong young cat and old cat. |
| Transformer | Walking through the storm. | See it, fearless in the face of any difficulties, how amazing! | My brain ran out of battery. | Smoking makes you age faster. |
| MEMEIFY | I walk, the wind blows, so I'm cool. | Who said only fish can be so free in the water? Look at this dog. | Don't know what I'm thinking about. | See the change with every cigarette. |
| BLIP-2-7B | The white Pomeranian's fur flutters in the wind. | Dog and duck walking on the beach. | A man puts water bottles next to each other. | two pictures of tom and jerry, one with a cat and the other with a dog. |
| MiniGPT-4-7B | It's a white dog strolling down the streets. | Dogs and ducks have fun in the sand! | It looked like he was going to be drowned by his own thoughts. | I don't know what the image is. |
| InstructBLIP-7B | This dog is so fluffy, it's like a cloud walking on four legs. | This is what happens when you let your dog walk the ducks. | Water, water everywhere, but not a drop to drink. | Tom: 'I'm gonna get you, Jerry!' Jerry: 'I'm not scared of you, Tom.' |
| LLaVA-7B | Wow, my fur is amazing! | Dogs are walking along the beach, while ducks are birds. | I'm not thirsty, I'm just looking at the water bottle. | When you're trying to quit smoking but still need that nicotine fix. |
| Unified-IOXL-2B | Two hairy dogs. | Dog and a group of ducks. | I am staring at two bottles. | A powerful cat and a cat on crutches. |
| Shikra-7B | Does this look like you? | A dog and ducks walk on the beach. | I am thirsity and I need two bottles of water. | What the cat looks like after smoking. |
| Qwen-VL-Chat-7B | Dog: It's too hard for me, the wind is so strong that I'm out of shape. | What a strange combination of dogs and ducks walking on the beach. | I need two bottles of water: one to drink and the other to prove that I'm not thirsty. | Cats can't escape time. |
| LLaVA1.5-7B | This dog looks like a little cartoon character. | It's a duck and it's running on the beach. | He's a handsome guy and looks funny. | This cat looks like it smokes, and it becomes like this when it smokes half of it. |
| GPT4v | When you accidentally open the front camera. | When you lied on your resume about being a great swimmer. | When you buy a bottle of water, only to find that you are already in the water. | When gym membership cards become the most expensive bookmarks. |
| **XMᴇCᴀᴘ** | **Who said people who resist the wind can't be graceful?** | **It tells us that in the face of difficulties, someone is always behind us.** | **I'm thinking, what should I have for dinner?** | **Do you still think tobacco is your friend?** |

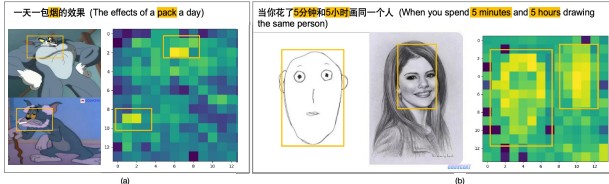

**Figure 4: Illustrative interpretation of meme-caption associations for more multi-image memes.**

not using cropping and rotation operations provided by AutoAugment. This absence may affect the model's ability to extract deep features from the image, such as the morphology of objects and color gradients; W/o TA, Without Text Augmentation, indicates that text augmentation techniques, specifically back-translation, are not used in processing textual data. Back-translation is employed to increase the diversity of textual data and assist the model in extracting deep features from both the original text and the back-translated text; W/o COH, Without Chain of Humor, means that the "Chain of Humor" template, inspired by the "chain of thought" approach, is not used in the process of generating meme captions. This template helps the model to construct structured texts, involving core concepts, emotions, events, consequences, and humor elements. In fact, we also try methods similar to simple CoT, such as: i) "Let's think outside the box. Please read the picture carefully and write a surprising and funny caption." ii) "Let's think outside the box. Please read the picture carefully and write a surprising and funny caption. Try to go wild and use your associative imagination. The more creative, the better." iii) "Let's think outside the box. Please

read the picture carefully and write a surprising and funny caption. Please use associative imagination based on the object in the image." These prompts do not perform as well as the current CoH, possibly because CoH provides possible angles for XMeCap to generate humorous captions; W/o A, Without Attention Mechanism, implies that the attention mechanism based on the Transformer model is not utilized to calculate the correlation between image features and caption features. This mechanism finely maps specific regions of the image to words in the text; W/o CL, Without Contrastive Learning, signifies that contrastive learning methods are not used in model training. Contrastive learning enhances the model's distinguishing ability by contrasting positive samples (actual captions) with negative samples (captions from other images); W/o RL, Without Reinforcement Learning, indicates that reinforcement learning is not integrated into the model training to optimize meme caption generation. Reinforcement learning involves evaluating the quality of captions and adjusting generation strategies based on these evaluations to create captions more aligned with human preferences.

Among the components, the attention mechanism displays the most pronounced effect, underscoring the significance of cross-modal interaction. The performance when removing "chain-of-humor" further emphasizes the importance of its cross-modal bridges. Contrastive Learning effectively distinguishes the latent relationships between individual memes and their corresponding captions. However, the effects of image and text augmentation were relatively subdued, suggesting that mere alterations without changing the core content of images or captions may not substantially enhance performance.

### 4.5 Illustrative interpretation

Meme caption generation hinges on pinpointing textual cues that align with the image's core humor. A successful approach should emphasize key phrases in harmony with the image's salient features [12, 17, 39, 40]. We use blue and orange to highlight the corresponding text. In the single-image meme above, "integrate" and "expose" stand out. We adopt FLIP [31] to draw a heatmap, and find that "integrate" corresponds to the yellow-highlighted region, representing a black dog among black ducks. Conversely, "expose" relates to the blue-highlighted area, marking a distinctive white feather on the dog's nose. This meme captures the humor of an effort to fit in yet unintentionally standing out. For the multi-image meme below, "different" and "greeting" are focal. The heatmap pinpoints two areas: a handshake and raised eyebrows. Although distinct, both gestures signify greetings, highlighting varied forms of acknowledgment. Our framework also highlights key parts in more multi-image memes such as the "head and cigarette" in both two subimages in the third meme (Fig. 4(a)), and the "simple face" in the left subimage and the "detailed face" in the right subimage in the fourth meme (Fig. 4(b)) with orange boxes for generating humorous captions.

### 4.6 Case Study

The case study in Table 9 and Table 10 indicates that our XMeCap closely matches GPT4's performance, notably in negative categories (self-mockery and mock others). Compared to baselines like s2s, Dank Learning, and transformer, our approach surpasses, showing

a deeper grasp of humor nuances. However, the XMeCap needs enhancement in the positive categories (self-praise and praise others), suggesting further refinements. Error analysis points to challenges in accurate caption generation. Context discrepancies can dilute humor. Although XMeCap is precise in image description, it sometimes miss humor or creativity. There's also the issue of cultural sensitivity and potential offensiveness. Difficulties with multi-image memes emphasize the need for improved captioning techniques.

## 5 RELATED WORK

Multi-modal humor research is expanding. Wu et al. [56] explore TV-sitcom dialogues, while Patro et al. [47] focus on 'Big Bang Theory' for humor detection. Chauhan et al. [8] and Li et al. [32] provide humor datasets from TV and memes respectively. Devillers et al. [25] consider laughter in robot interactions. Chen and Jiang [10] and Tsakona [52] address humor theories. Hasan et al. [26] and Yang et al. [57] present models for multimodal humor understanding and labeling. Alnajjar et al. [1] and Chauhan et al. [7] delve into TV shows and sentiment in humor. Regarding humor generation, Ritschel et al. [49] and Ritschel et al. [48] focus on robot humor. In meme generation, Sadasivam et al. [50] to Wang et al. [54] offer tools, datasets, and systems for memes. However, distinguishing between single and multi-image memes hasn't been a focus until our research, which offers a distinct approach for both single-image and multi-image memes.

Textual humor generation seeks to produce comedic content. Templates often involve lexical changes via tools like WordNet, as demonstrated by Sjöbergh and Araki [51] in Japanese comedy and Hong and Ong [28] for puns. However, they can be formulaic. Conversely, neural models promise more originality. For example, works by Li et al. [33] and Yu et al. [58] use such models for pun creation. However, humor isn't solely textual; images play a role, which our research emphasizes.

## 6 CONCLUSIONS

Humor poses a great challenge for human-machine interaction. This study has illuminated the intricate dynamics of humor in the multi-modal realm of memes, emphasizing on the impact of multi-images on meme captioning. Through our innovative XMeCap framework, we have established a deeper comprehension of meme image-text relationships, achieving state-of-the-art performance in meme caption generation as well as multi-modal humor detection. As we advance, it's imperative to explore the cross-cultural adaptability of our method, understanding the subtle variations in humor across different societies. Additionally, integrating more sophisticated semantic analysis tools could further refine the quality of generated captions.

## ACKNOWLEDGEMENTS

This work is supported by Science and Technology Commission of Shanghai Municipality Grant (No. 22511105902), the National Natural Science Foundation of China (No.62072323), Shanghai Science and Technology Innovation Action Plan (No.22511104700).

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
