# OpenReview forum: "XMeCap: Meme Caption Generation with Sub-Image Adaptability"
_acmmm.org/ACMMM/2024/Conference — MM2024 Poster_

### Official Review · Reviewer_U5c5 · 2024-05-06

**Rating:** 3
**Confidence:** 3

**Summary:**

This study has illuminated the intricate dynamics of humor in the multi-modal realm of memes, emphasizing on the impact of multi-images on meme captioning. The innovative XMECAP framework achieved state-of-the-art performance in meme caption generation as well as multi-modal humor detection.

**Strengths:**

1. The XMECAP methodology features a novel reward model that integrates both global and local similarities between visuals and text through supervised fine-tuning and reinforcement learning for meme caption generation.

2. Superior Performance Validated: Extensive experiments have confirmed the superiority of the XMECAP approach over current benchmarks in both single and multi-image meme captioning, along with promising results in conventional multi-modal humor detection tasks.

3. The research underscores the ability of the proposed XMECAP to discern intricate associations between memes and their corresponding captions through visualization analysis, paving the way for future advanced meme-related research.

**Limitations:**

1. In Table 1, the first row, I assume it includes data from all four categories, yet the max/min tokens are 24 and 6 respectively. The max tokens in captions of praise others in memes is 25, and the min tokens in captions of self-mockery in memes is 5. This seems to contradict my understanding.

2. Section 3.2 seems to be confusing and does not correspond with Figure 1. Specifically, as follows:
- The "shared projection layer" in the middle of Figure 1 is described at the beginning of Section 3.2 as two trainable linear layers, and I am not clear about the meaning of 'shared'.
- In line 268, "self-attention mechanism of the LLM," it appears to me to be just a regular self-attention mechanism, and I'm not clear on the significance of emphasizing LLM.
- The symbols for global attention and token-level attention seem to contradict Figure 1, which makes it difficult for me to understand.
- In the last paragraph, there are two instances of  $S_{I_i,j}$, and I suspect there might be a typographical error with one of them.

3. Lines 367 and 414 describe these parameters as trainable weights (with a singular/plural grammatical error in line 367), yet in lines 427 and 430, these parameters are set to specific numerical values, which is confusing to me. Assuming these parameters are indeed trainable weights, not constraining them could be fatal because the model might rely on these trainable weights to achieve a falsely minimized loss.

4. There is a lack of description for the baselines, and it is unclear whether all baselines have been fine-tuned on this dataset.

5. Ablation studies should differentiate between single-image and multi-image scenarios, as I am more interested in the impact of each component on the performance of multi-image settings.

6. For Section 4.5, more examples should be presented to comprehensively illustrate the role of the proposed method.

7. In line 453, "Detailed" should be in lowercase as it is not the beginning of a sentence. Additionally, the author refers to Appendix A, which I have not seen, and to my knowledge, ACM MM prohibits the use of appendices.

**Suitability:**

3

---

### Official Review · Reviewer_Zdyg · 2024-05-24

**Rating:** 5
**Confidence:** 4

**Summary:**

This paper presents a novel framework for meme caption generation that focuses on handling multi-image memes. The framework employs supervised fine-tuning and reinforcement learning with an innovative reward model considering both global and local similarities between visuals and text. The framework aims to address the unique challenges posed by multi-image memes, such as integrating composite information and maintaining consistency across shared captions. The authors constructed a new meme dataset, categorized by structure (single-image and multi-image) and emotion (self-praise, praise of others, self-mockery, and mockery of others), to train and evaluate the model. The experimental results demonstrate significant improvements in caption generation for both single and multi-image memes compared to existing benchmarks.

**Strengths:**

Theoretical Approach: The paper provides a robust theoretical foundation for the proposed framework. The use of adaptive transformation layers and attention-guided text generation is well-motivated, and the detailed explanation of the reward model enhances the understanding of the approach.

Comprehensive Dataset: The construction of a large-scale, well-balanced meme dataset, categorized by structure and emotion, is a notable contribution. This dataset enables thorough evaluation and facilitates future research in meme caption generation.

Evaluation and Results: Extensive experiments validate the superiority of the XMeCap framework over current benchmarks. The performance improvements in both single and multi-image meme caption generation are well-documented, with detailed comparisons to various baselines.

Clarity: This paper is well-written and easy-to-follow with detailed explanation and interpretable figures.

**Limitations:**

Language Limitation: The dataset and experiments are focused on Chinese memes, which might limit the generalizability of the results. While the authors use translation tools to mitigate this, the nuances of humor in different languages may not be fully captured.

Evaluation Metrics: The evaluation relies heavily on human judgment for metrics like informativeness, relevance, creativity, and humor. Although inter-rater agreement is calculated, the subjective nature of humor might introduce variability in the evaluation results.

**Suitability:**

3

---

### Official Review · Reviewer_8ygc · 2024-05-25

**Rating:** 5
**Confidence:** 3

**Summary:**

This paper breaks down memes into multiple images for better image captioning and introduces XMECAP to combine reinforcement learning and supervised learning for encouraging similarity between text and images, similar to CLIP pre-training, but supervised. They also construct their own dataset containing 12,320 Chinese memes from social media.

**Strengths:**

1. Breaking down memes into multiple images is necessary in this evolving digital landscape where memes are increasingly creative and may have multiple meanings.
2. The comparison with GPT-4 proves that XMECAP is at least up-to-par with SoTA models.
3. A wide variety of metrics are used to evaluate the generated captions.
4. Table 8 and 9 prove how contemporary models may generate misleading or opposite captions. XMECAP has the upper hand here.

**Limitations:**

1. Using LLMs (and VLMs) to decode memes is intuitive but very computationally expensive for real-world usage and content moderation. This framework could partially benefit from distillation.
2. Human evaluation may include more criteria: for example, sarcasm, irony, etc. which are particularly challenging for vision-language models (although it slightly coincides with creativity).
3. For LLM prompting, it would be good to experiment with other templates besides the Chain-of-Humour.

**Suitability:**

3

---

### Official Review · Reviewer_TwN1 · 2024-05-28

**Rating:** 4
**Confidence:** 2

**Summary:**

This paper reports on a system for generating humorous captions for a sequence of two or more images.  It claims to improve performance, as measured by several human and machine evaluations, by about 5%. The system also can be used for single images, with an improvement of about 4%.  The system relies on a custom-created database, as well as the ground truth provided for it by three human annotators.

**Strengths:**

What is new: a new combination of off-the-shelf methods: for finding subimages, for creating phrases for them, and for stitching the language results together according to a "humor grammar".  The contribution appears to be that of a particular synthesis, rather than any new insights or methods, as indicated by the marginal improvement in performance.  The database appears more interesting, given its size (18K memes) and four-part annotation (praise/mock x self/others).

The experimentation and evaluation are well done (Tables 4, 5), and believable examples are presented (Table 8, 9), perhaps even somewhat to excess.

**Limitations:**

The Abstract is generic and over-enthusiastic ("a new frontier"?!).  It needs some quantification about approach, datasets, competitors, and performance.  In fact, the reader is surprised to discover later in the paper how extensive the evaluations were, even if the results appear modest.

The Figure 1 multi-image has only *three* subimages, but the text talks about *four*.of them (on text line 54).

It never becomes clear why this particular system is a contribution: the three contributions listed at the end of the Introduction don't say anything unique to this paper: "novel methodology", "extensive experiments", "intricate associations"?  If there is anything that may be truly original, it might be the reward model, Equation 13, which gets its strength from the sequential ordering of captions based on human ground truth.

The ablation studies in Table 5 seem to suggest that attention is far more important than "chain of humor".  But this is not explained, and it seems to suggest that attention to detail is more important for humor than a global overview, somewhat contradicting one of the proposed virtues of the system.

The Conclusions don't really say anything and can be omitted.

**Suitability:**

3

---

### Meta-Review · Area_Chair_LVu1 · 2024-06-30

**Recommendation:** Accept (Poster)
**Confidence:** 5

**Metareview:**

The paper presents XMECAP, a system designed to generate humorous captions for sequences of images, achieving modest performance improvements over existing methods. The key strengths include the innovative combination of existing techniques for sub image identification, phrase generation, and stitching results using a "humor grammar," as well as a large, well-annotated meme database. The experimentation and evaluation are thorough, with believable examples provided. However, the contributions lack clear novelty, relying on existing methods rather than new insights, and the abstract needs to be more specific about the approach, datasets, and performance. The explanation of the system's unique contributions is vague, and the ablation studies suggest the importance of attention over the "chain of humor," which is not well-explained. To address these concerns, the authors should clarify the unique contributions of their system, provide a detailed analysis of the attention mechanism versus the "chain of humor," and enhance the abstract with quantifiable details. Additionally, improving the clarity and consistency of the figures and descriptions, expanding the human evaluation criteria, and exploring alternative LLM prompting templates would strengthen the paper for the camera-ready version.